https://doi.org/10.1038/s41467-019-08430-8　　**OPEN**

# Catalytic inverse vulcanization

Xiaofeng Wu [1], Jessica A. Smith [1], Samuel Petcher [1], Bowen Zhang [1], Douglas J. Parker [1], John M. Griffin [2] & Tom Hasell [1]

The discovery of inverse vulcanization has allowed stable polymers to be made from elemental sulfur, an unwanted by-product of the petrochemicals industry. However, further development of both the chemistry and applications is handicapped by the restricted choice of cross-linkers and the elevated temperatures required for polymerisation. Here we report the catalysis of inverse vulcanization reactions. This catalytic method is effective for a wide range of crosslinkers reduces the required reaction temperature and reaction time, prevents harmful $H_2S$ production, increases yield, improves properties, and allows crosslinkers that would be otherwise unreactive to be used. Thus, inverse vulcanization becomes more widely applicable, efficient, eco-friendly and productive than the previous routes, not only broadening the fundamental chemistry itself, but also opening the door for the industrialization and broad application of these fascinating materials.

[1] Department of Chemistry, University of Liverpool, Crown Street, Liverpool L69 7ZD, UK. [2] Department of Chemistry, Lancaster University, Lancaster LA1 4YB, UK. Correspondence and requests for materials should be addressed to T.H. (email: T.Hasell@liverpool.ac.uk)

In modern society, synthetic polymers are ubiquitous to human life and are among the most extensively manufactured materials on earth. There are now around 380 million tonnes of plastic produced annually[1]. The environmental impact and sustainability of any alternative synthetic polymer is therefore important to consider, and should ideally align with the principles of green chemistry[2]. However, the vast majority of synthetic polymers are produced from limited resources derived from petrochemicals[3]. There is therefore a significant challenge in materials chemistry to identify sustainable building blocks that provide monomers generated from renewable biomass, re-purposed agricultural, or industrial waste[4,5].

Elemental sulfur is readily available and inexpensive, being produced in excess of 70 million tonnes each year as an unwanted by-product of petroleum refining and gas reserves[6]. Sulfur widely used for the production of commodity chemicals, such as sulfuric acid, fertilizers, and vulcanization of natural and synthetic rubbers. Despite this, supply greatly outweighs demand, creating large unwanted stockpiles and a global issue in the petrochemical industry known as the "excess sulfur problem". The problem will grow in scale as demand for energy pushes the need to use more sulfur-contaminated sour petroleum feed-stocks. From this perspective, there is interest in exploiting this un-tapped, low-cost sulfur for materials[7–12].

Although sulfur can be polymerized in a pure form (Fig. 1a), the resultant polymers are not stable and readily depolymerize to $S_8$. The recent discovery of inverse vulcanization, which uses organic crosslinkers to stabilize the sulfur chains, has heralded a class of materials pioneered by Pyun and Char in 2013[8]. These materials are made predominantly from elemental sulfur without the need for harmful organic solvents. Molten sulfur acts as the reaction solvent itself, as well as monomer and initiator during the molten stage. The growing high sulfur polymers, henceforth referred to as thiopolymers, are stabilized against depolymerization by reaction with an organic cross-linker. The synthetic process is simple, scalable, and highly atom efficient—an excellent example of green chemistry. As well as merely substituting for carbon based polymers, polymers made from sulfur have the potential for radically different properties, enabling unique applications. For example, the optical properties of these polymers are quite different to those of carbon based polymers, which have a refractive index typically in the 1.5–1.6 range, and poor transparency to near infrared light. Conversely, thiopolymers have refractive indices as high as 1.86, and high infrared transparency, giving excellent properties as lenses and in thermal imaging applications[13–15]. The low cost of these materials also gives them excellent potential for bulk construction applications derived from their high thermal[16] and electrical insulating properties. Despite their crosslinked structure, the reversibility of sulfur-sulfur bonds gives vitrimer[17] behavior, allowing recycling[18], and repair[14]. Other already reported applications include LiS batteries[8,19,20], water purification[10,21–25], the stabilization of metal nanoparticles and quantum dots[26–29], and antimicrobial materials[30], and there are doubtless many more applications yet to be discovered.

Various cross-linkers, such as those shown in Fig. 1b, have been reported to form sustainable polymers[31]. Crosslinkers used include industrial feed-stocks, such as diisopropylbenzene (DIB)[8], divinylbenzene (DVB)[19], and dicyclopentadiene (DCPD)[32], as well as renewable sources such as limonene[21], vegetable oil[33,34], myrcene[32], and diallyl disulfide[35]. In general, the reactions require heating to over 160 °C to induce thiopolymerization. Some reported reactions require even harsher conditions of 180 °C or more[8,19,21,35,36]. For other co-monomers, such as styrene, 130 °C is enough to form oligomeric material[37]. Conventional sulfur-olefin reactions are characterized as low

temperature reactions up to about 140 °C, and high temperature reactions above 140 °C[38]. Avoiding higher temperatures (over 140 °C) is crucial in minimizing the formation of hydrogen sulfide, thiols, and dehydrogenation of olefins during vulcanization[39]. A catalytic pathway that lowered the required temperature for inverse vulcanization is therefore highly desirable for its safe scale-up, by allowing $H_2S$ production[40] to be reduced. Lower reaction temperatures are also likely to help avoid dangerous auto acceleration of the reactions by the Trommsdorff-Norrish effect[41], that can occur during inverse vulcanization (see supplementary figure 1)[32]. In addition, there are many cross-linkers prohibited from polymerization with sulfur by their lack of reactivity and/or low boiling point.

We report here the investigation of catalytic inverse vulcanization. This catalysis enables a series of polymers, (Fig. 1c, d), as well as improving the reaction and properties of existing thiopolymers. This catalytic process also significantly reduces highly toxic $H_2S$ generation from the reaction, which will be a critical issue for industrial application.

## Results

**Screening of catalysts**. When screening potential crosslinkers, some were found to be un-reactive to sulfur even over 200 °C. Inspired by accelerators used in conventional vulcanization[42], the introduction of catalysts into this inverse vulcanization was trialed (Fig. 1e; Table 1). The reaction of cross-linker ethylene glycol dimethacrylate (EGDMA) with sulfur was used as a model reaction, as it was found to be un-reactive without catalysis.

Sulfur, by itself, is a slow vulcanizing agent, requiring high temperatures and long heating periods[42]. Metal salts, oxides and complexes have been successfully applied as accelerators for conventional vulcanization[42], ZnO being one of the most commonly used. From our tests, ZnO did not show catalytic activity for this inverse vulcanization reaction, nor did inorganic complexes from copper, zinc, or iron chloride (Entries 1–6, Table 1, supplementary figure 2). Zinc stearate[43] did show some catalytic activity, but unfortunately even after curing there was unreacted or depolymerized sulfur. Replacing the stearate ligand with diethyldithiocarbamate (DTC), the reaction becomes noticeably quicker, with the color changing from yellow to orange-red within minutes after the addition of cross-linker, later becoming a homogeneous rose-red clear solution and finally a deep-red viscous gel, seizing the stirrer bar. In contrast, the same reaction without this catalyst affords two separated layers with minimal reaction even up to 200 °C. The catalyzed product is a hard black solid, insoluble even in strong organic solvents such as tetrahydrofuran and chloroform. Solid state nuclear magnetic resonance spectroscopy (NMR) shows evidence of C–S bond formation, and loss of C=C bonding, as does Infrared spectroscopy (supplementary figures 3 and 4). The DTC ligand seems to be more crucial than the metal, as other metals such as Fe and Cu were found to also work effectively with this ligand. NaDTC notably reacted the quickest with EGDMA, forming a gel in only a few minutes. However, this short mixing time may lead to inhomogeneous products and NaDTC was not compatible with all crosslinkers. $Zn(DTC)_2$ is known to be an effective accelerator in conventional vulcanization and showed activity for a broad range of crosslinkers, and was therefore selected for further optimization. It is important to note the viability of metals such as Fe, Co, or Cu which may be preferable in terms of cost or safety. With several metal complexes showing viability, it seemed possible the catalytic effect could arise from simply the DTC ligand itself, rather than the metal, by a process similar to reversible addition−fragmentation chain-transfer polymerization (RAFT)[44]. To test this, thiram (effectively DTC-DTC) and a

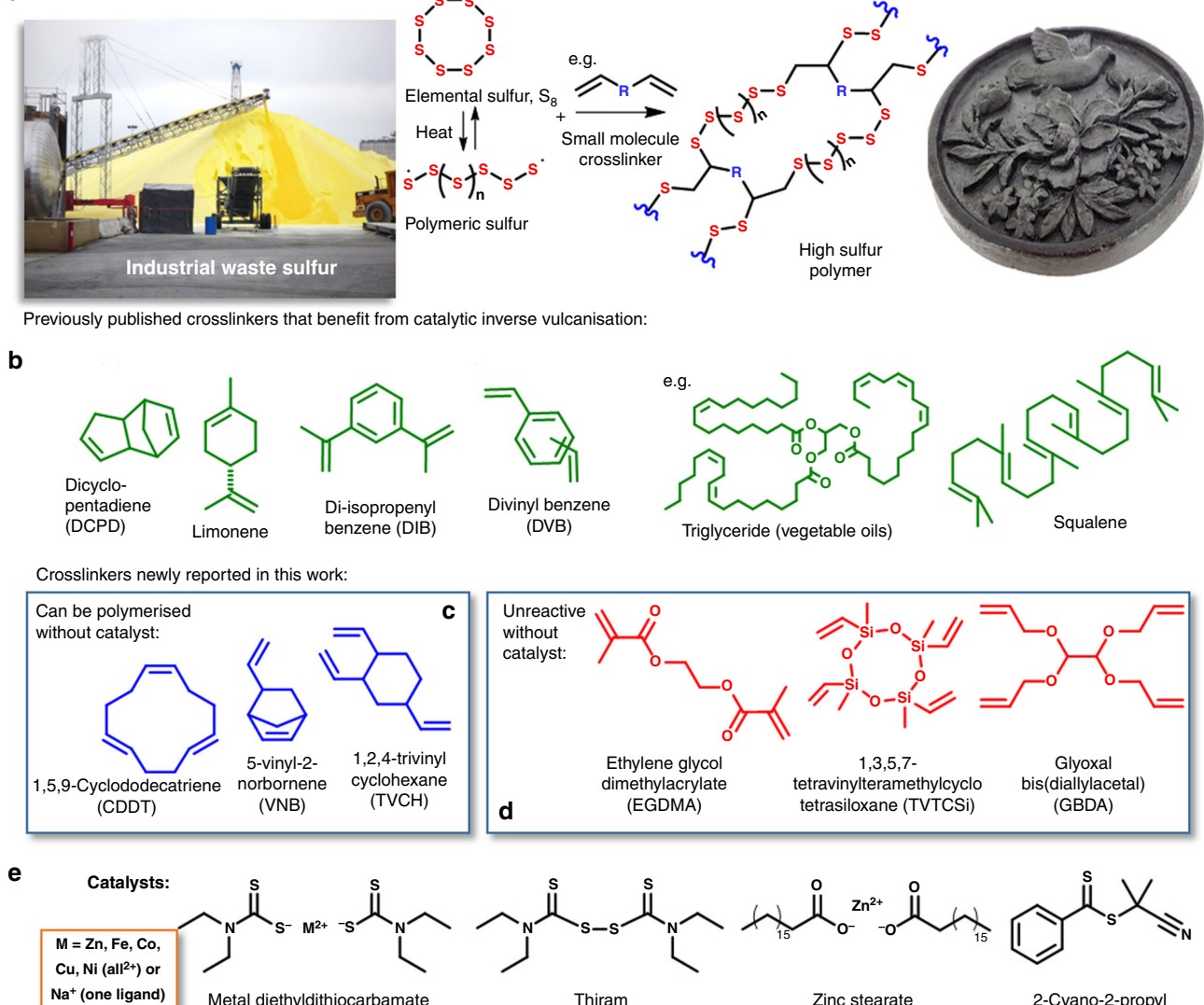

**Fig. 1** Synthesis of polymers from elemental sulfur and organic crosslinkers. **a** A generalized reaction scheme for the inverse vulcanization of sulfur, where R indicates the core of an unsaturated small molecule crosslinker. The crosslinker reacts with the polymerizing sulfur to stabilize it against depolymerization, thus allowing industrial waste elemental sulfur (left, Alamy stock photo) to be converted to stable polymeric objects (photo right). **b** A wide range of crosslinkers can be used for inverse vulcanization, the structures shown here in green are a diverse range of previously published crosslinkers we chose to test for catalyzed polymerization. **c**, **d** Crosslinkers for inverse vulcanization that have not been previously reported and that are either reactive (in blue, **c**) or unreactive (in red, **d**) without catalysts. **e** Catalysts trialed for inverse vulcanization

common RAFT agent (2-Cyano-2-propyl benzodithioate) were also trialed, but showed poor and no activity, respectively.

**Effects of catalysis**. As well as allowing previously unreactive EGDMA to be polymerized with sulfur, the Zn-DTC catalyst was also tested for a range of other crosslinkers both previously reported, and untested (Fig. 1b–d, and supplementary figures 3–14, supplementary tables 1–3). All catalyzed reactions formed solid polymers that could be molded into objects (Fig. 2a), and that were thermally stable to 200 °C (supplementary figures 15–24).

**Prevention of H₂S production during polymerization**. The generation of toxic $H_2S$ gas as a by-product has been noted for some inverse vulcanization reactions[27,40,45]. To test this, reactions were performed both with and without catalyst, with temperatures chosen to achieve comparable rates of reaction, and the volume of gas produced was measured. Catalyzed reactions were found to produce up to seven times less, down to negligible levels (Fig. 2b). This is likely the result of the lower temperatures needed, as higher temperatures are known to produce $H_2S$ and thiols in conventional vulcanization[39], but may also stem from differences to the reaction mechanism itself; reactions of sulfur with limonene produced significantly less $H_2S$ in the presence of a catalyst, even when performed at the same temperatures (supplementary figure 25).

**Unlocking alternative crosslinkers**. A key benefit of catalytic inverse vulcanization, is to bring unreactive cross-linkers into use, thus expanding the range of possible S-rich polymers. Along with EGDMA, glyoxal bis(diallylacetate) (GBDA) and 1,3,5,7-tetra-vinyltetramethylcyclotetrasiloxane (TVTCSi) crosslinkers also

**Table 1 Screening of catalysts for inverse vulcanization of sulfur with EGDMA**

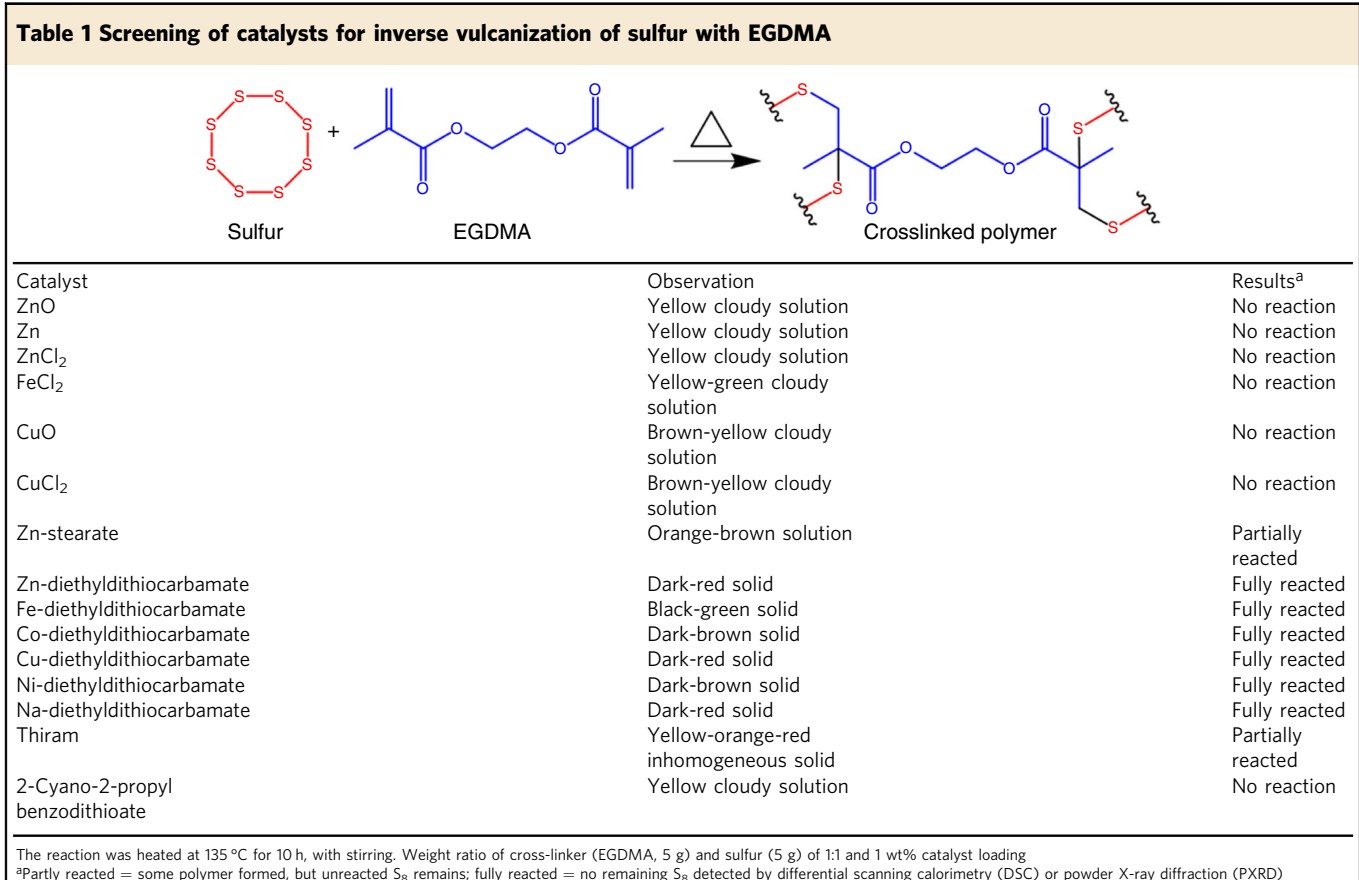

| Catalyst | Observation | Results[a] |
|---|---|---|
| ZnO | Yellow cloudy solution | No reaction |
| Zn | Yellow cloudy solution | No reaction |
| ZnCl$_2$ | Yellow cloudy solution | No reaction |
| FeCl$_2$ | Yellow-green cloudy solution | No reaction |
| CuO | Brown-yellow cloudy solution | No reaction |
| CuCl$_2$ | Brown-yellow cloudy solution | No reaction |
| Zn-stearate | Orange-brown solution | Partially reacted |
| Zn-diethyldithiocarbamate | Dark-red solid | Fully reacted |
| Fe-diethyldithiocarbamate | Black-green solid | Fully reacted |
| Co-diethyldithiocarbamate | Dark-brown solid | Fully reacted |
| Cu-diethyldithiocarbamate | Dark-red solid | Fully reacted |
| Ni-diethyldithiocarbamate | Dark-brown solid | Fully reacted |
| Na-diethyldithiocarbamate | Dark-red solid | Fully reacted |
| Thiram | Yellow-orange-red inhomogeneous solid | Partially reacted |
| 2-Cyano-2-propyl benzodithioate | Yellow cloudy solution | No reaction |

The reaction was heated at 135 °C for 10 h, with stirring. Weight ratio of cross-linker (EGDMA, 5 g) and sulfur (5 g) of 1:1 and 1 wt% catalyst loading
[a]Partly reacted = some polymer formed, but unreacted S$_8$ remains; fully reacted = no remaining S$_8$ detected by differential scanning calorimetry (DSC) or powder X-ray diffraction (PXRD)

only reacted viably with sulfur in the presence of catalysts. Monomeric sulfur, S$_8$, readily crystallizes, and therefore if the polymerization is not complete, or depolymerization occurs, the presence of S$_8$ crystals can be detected in differential scanning calorimetry (DSC) (Fig. 2c and supplementary tables 1 and 2), and by powder x-ray diffraction (PXRD) (Fig. 2d and supplementary table 3). The crosslinkers containing heteroatoms (Fig. 1d) all showed residual sulfur in the absence of catalyst, but complete reaction with catalyst. It is likely that the heteroatoms of these crosslinkers deactivate the vinylic positions. Unlocking the reactivity of acrylates, which do not react with sulfur on their own[37], opens up many alternative potential crosslinkers. Accessing more crosslinkers is useful not only for additional polymer themselves, but also as co-monomers for blends, to control properties. For example, blending different crosslinkers with TVTCSi can produce stable polymers ranging from glassy solids with no detectable $T_g$ (with TCDD or DCPD), to rubbery solids with sub-room temperature $T_g$ (with EGDMA, 8.7 °C; or Farnesol, 4.3 °C, see supplementary table 4).

**Increased rate of reaction**. The catalyzed reactions typically require significantly less time to reach completion (Fig. 2e). Reducing the required reaction times and temperature for these polymers is significant if they are to be scaled up for the bulk applications that are allowed for by the low cost, availability, and renewability of many of the feedstocks. If the catalyst loading is varied there is a clear trend of reduced time with increasing catalyst addition (Fig. 2e inset). In open reactions there is a gradual loss of mass by evaporation of the monomers. The increased reaction rate therefore also corresponds to a higher yield (Fig. 2f, and supplementary figure 26).

**Improved properties**. Of the crosslinkers able to react in the absence of catalyst (Fig. 1b, c), as well as reduced reaction times, many also showed an increase in glass transition temperature ($T_g$) when catalyzed (Fig. 2g, S27–S33, supplementary table 5). It is likely the catalysis produces more crosslinking, and a more even distribution of sulfur leading to shorter sulfur chains between crosslinkers. Sulfur-DCPD copolymers show a particularly pronounced difference in properties between catalyzed and uncatalysed reactions—with an increase in $T_g$ from 38 to 89 °C, (supplementary figure 17). This behavior in DCPD is linked to the difference in reactivity between the double bonds. At temperatures below 140 °C, only the norbornene double bond is reactive[32,38], with higher temperatures needed to activate the cyclopentene bond. Catalysis allows both bonds to react in the low temperature regime.

Limonene is desirable as a crosslinker for sulfur as it is a renewable by-product of the citrus industry, and its thiopolymer has been shown to have potential for mercury capture[21]. As well as increasing the $T_g$, catalysis also improved the shape persistency of sulfur-limonene, giving it a reduced tendency to creep (supplementary figures 34–36). The low molecular weight of sulfur-limonene copolymers (<1000 $M_w$ by gel permeation chromatography, supplementary figure 36) in comparison to other inverse vulcanized polymers provides solubility in organic solvents. This solubility was used to coat commercial silica gel with sulfur-limonene copolymers, in order to test their function as a filtration medium for mercury. After coating with a 10 wt% loading of polymer, the silica gel was a fine free flowing powder, maintaining the same particle size, and without aggregation of the particles (photographs and SEM images shown in supplementary figure 38). Exposure of this powder to aqueous solutions of mercury chloride gives a significant increase in mercury uptake

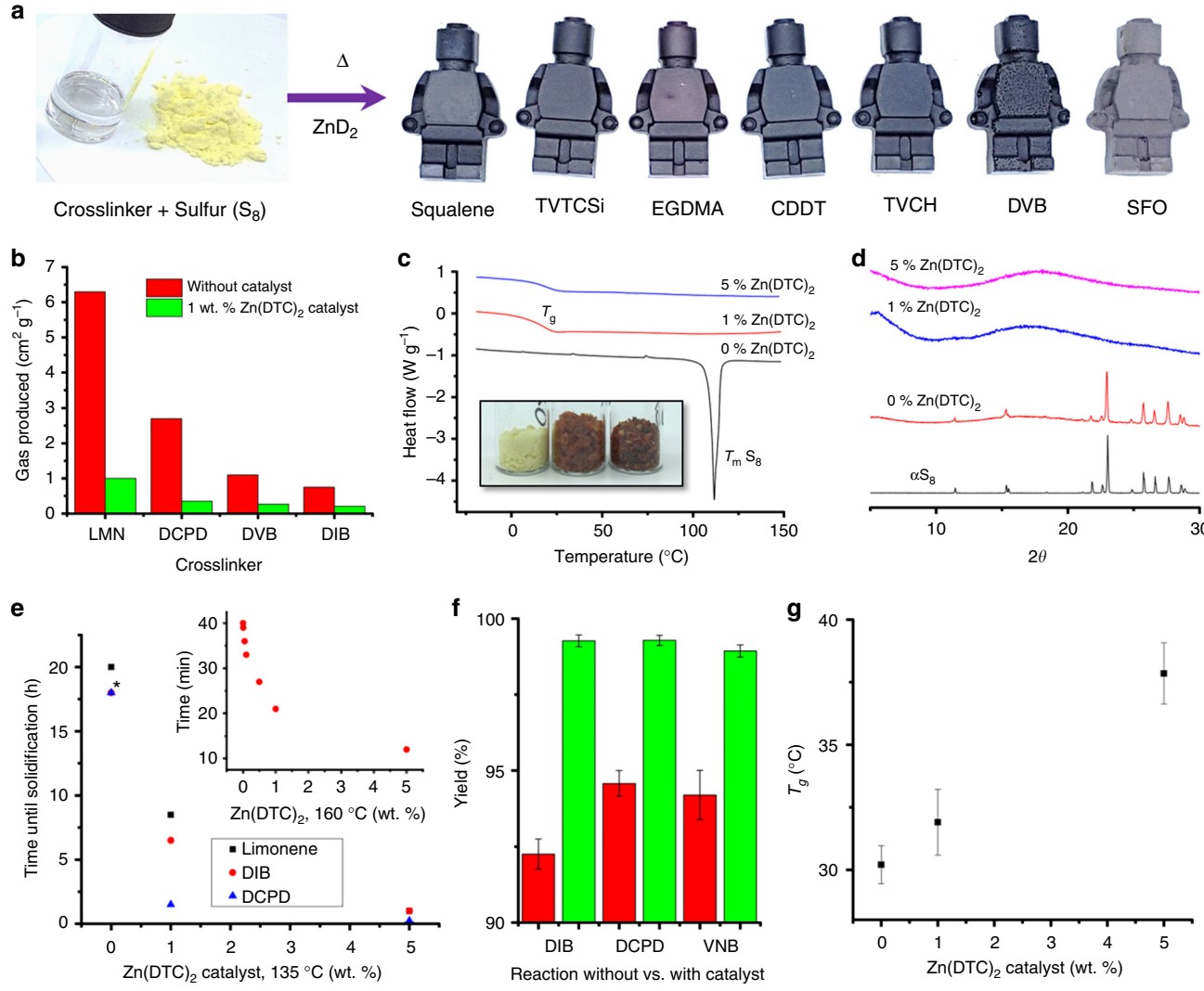

**Fig. 2** Characterization of catalyzed thiopolymers. **a** Photographs of a crosslinker (EGDMA) and elemental sulfur and examples of molded objects of catalyzed thiopolymers. **b** Volume of gas produced during reactions of sulfur with crosslinkers, with (red) and without (green) catalysts. **c** Offset DSC traces of sulfur reacted with EGDMA. In the absence of catalyst, the melting of $S_8$ crystals at ~120 °C is apparent. Inset photograph shows the color of the products (from left to right: 0, 1, and 5 wt% catalyst). **d** Offset PXRD patterns of sulfur reacted with EGDMA, in the absence of catalyst the diffraction of $S_8$ crystals is apparent. **e** Reaction time plotted against catalyst loading at 135 °C. Asterisk (*) indicate uncatalyzed DIB and DCPD took between 12 and 24 h (unobserved), plotted as 18 h. **f** The yield of open reactions performed at 135 °C, with (red) or without (green) 1 wt% $Zn(DTC)_2$ catalyst. Error bars given for standard deviation of 3 repeats. **g** Glass transition temperature of insoluble sulfur-squalene polymers as a function of $Zn(DTC)_2$ catalyst loading, plotted as the average of three parallel reactions

(Fig. 3a) in comparison to uncoated silica gel, which had negligible effect on mercury concentration (supplementary figure 39). As well as taking up mercury, crucial for environmental applications, the polymers also show affinity for removing gold from solution, relevant to mining and recovery applications[46]. Importantly, the high uptake is specific to heavy metals such as Hg and Au, with much lower uptake for other common metals (Fig. 3a and supplementary figures 40 and 41). The metal uptake increases with catalyst loading, potentially resulting from improved dispersion and bonding of sulfur, but with a possible contribution from the catalyst itself, which has been reported to bind metals[47]. A rapid uptake occurs immediately on exposure to mercury solution, followed by a more gradual uptake reaching equilibrium in a few hours (Fig. 3b). The isotherm has a steep uptake at low concentration, of most relevance industrially (Fig. 3c, and supplementary figure 42). The maximum capacity of 65 mg g$^{-1}$ sorbent corresponds to 716 mg of Hg per gram of

polymer—to our knowledge the highest uptake reported to date for inverse vulcanization.

**Mechanism**. Despite a long history of use, the mechanism of even uncatalyzed conventional vulcanization is not fully understood, and remains complex, difficult to characterize, and controversial[42]. Conventional vulcanization has been ascribed to either radical or ionic pathways according to homolytic, or heterolytic fission of $S_8$ rings (supplementary figure 43)[39,42,48], and even recently as initially radical, with ionic species generated after reaction of sulfur with organic species[49]. That said, the most widely agreed pathway for conventional vulcanization is via hydrogen abstraction of the α-position relative to the double bond, leading to a combination of crosslinking by proton substitution and addition across the double bonds, with substitutions of hydrogen for sulfur being the dominant factor (Fig. 4a)[39,42,48,50].

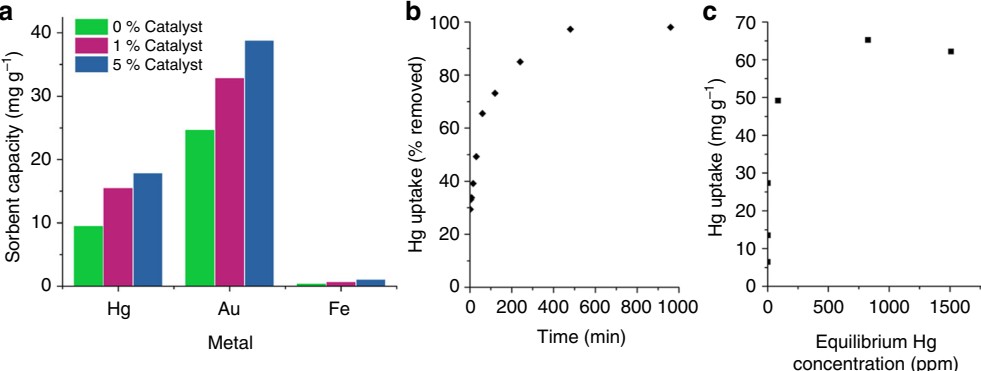

**Fig. 3** Metal uptake by thiopolymers. **a** Uptake of metal by sulfur-limonene coated silica gel from 400 ppm aqueous solution of mercury chloride and iron chloride, and 800 ppm gold chloride, with varying $Zn(DTC)_2$ catalyst loading, after 1 h. **b** The uptake of Hg from 1000 ppm $HgCl_2$ aqueous solution, by 5 wt % $Zn(DTC)_2$ catalyzed sulfur-limonene coated silica gel, as a function of time. **c** The uptake of Hg, by 5 wt% $Zn(DTC)_2$ catalyzed sulfur-limonene coated silica gel, as a function of equilibrium concentration, to determine maximum capacity

In comparison, inverse-vulcanization is relatively recent, and has yet to undergo as extensive an investigation into its mechanism. Most of the existing discussion describes inverse vulcanization as being bulk free radical copolymerization of unsaturated co-monomers in liquid sulfur[20], and invokes addition across the double bonds being either the only, or dominant feature (Fig. 4b)[8,10]. However, abstraction of hydrogen and $H_2S$ evolution have also been reported[40,51,52]. It is likely that both mechanisms, radical addition to the double bond, and hydrogen substitution, occur in both classes of vulcanization, with the ratio highly dependent on the temperature, as well as the proportion of sulfur.

The starting temperature of homolytic fission for $S_8$ has not been agreed, with reports ranging from 140 to 181 °C[49,53–56]. That catalysts allow temperatures below this range to be used may therefore make a crucial difference to the nature of the reaction. In the first report of inverse vulcanization, Pyun and co-workers reported that the polymerization of liquid sulfur above its floor temperature (159 °C, the temperature at pure sulfur exists mostly as polymers, rather than as $S_8$) was a key stage in the reaction (supplementary figure 43b)[8]. This is possibly the reason many un-catalyzed inverse vulcanizations are performed over 160 °C.

The α-proton of allyl groups is known to be very reactive and it has been proved thiyl radicals can abstract this α-proton atom during vulcanization[48,57]. Un-catalyzed polymerizations are likely to undergo a step-wise mechanism triggered by initial hydrogen abstraction, as in conventional vulcanization. The thiyl radicals abstract a proton first to generate carbon radicals on the C=C double bond, these carbon radicals will then initiate further polymerization. For catalytic inverse vulcanization we tentatively suggest the pathway shown in Fig. 4c. The metal-sulfur bond allows the opening of the $S_8$ ring at lower temperatures, and insertion of sulfur between the metal and DTC ligand to generate the active catalyst. The catalyst then brings the sulfur into proximity to the crosslinker, and lowers the energy barrier to bond formation. It is not clear whether this step is radical or ionic in character, and may be concerted. Repeated chain transfer and reaction will lead to highly crosslinked networks with more even distributions of sulfur. The lack of any activity shown by the conventional RAFT agent, 2-Cyano-2-propyl benzodithioate, results from the lack of S–S or metal-S bonds. These bonds are necessary for insertion of sulfur from the $S_8$ phase, and transport into the organic phase for reaction and catalysis. Thiram, with a reversible S-S bond, allows such a mechanism, but the efficiency was lower than for the metal-based catalysts. The metal coordinated form of the catalyst is likely more susceptible than

the thiram form to the insertion of sulfur oligomers into the catalyst. The more ionic nature of the metal-sulfur bond, in comparison to disulfide bonds, provides higher reactivity. Many inverse vulcanization reactions suffer from poor miscibility between the organic crosslinker and molten sulfur phases. The oleophilic and sulfur-philic moieties of $Zn(DTC)_2$ (Fig. 4d) allow it to act as an ideal phase transfer catalyst to shuttle reactive sulfur into the organic phase. When comparable complexes are used to increase the rate of reaction in conventional vulcanization, they are commonly referred to industrially as "accelerators" rather than catalysts. This mitigates both for the lack of complete understanding of the mechanism, and that it is not possible to separate and reclaim the active complex after polymerization, as it is incorporated in the product. Here we have used the term catalysis loosely for our process, for sake of accessibility, but acceleration may be more technically appropriate.

NMR analysis was performed of early stages of the polymerization of sulfur with DCPD, for which the forming oligomers are soluble (Fig. 5, supplementary table 6 and supplementary figures 44-49). Reactions were carried out in the low temperature regime (135 °C), with and without catalyst, and compared to high temperature reactions (initiated at 185 °C). When comparing the three conditions at equal reaction time (10 min, Fig. 5a), the catalyzed reaction has already begun to react, while there is no change for the uncatalyzed sample. The higher temperature reaction has progressed further for the same time; however, a significant degree of hydrogen substitution is evident, based on the appearance of peaks around 6.5 ppm. These peaks correspond to the norbornene C=C bond proton after α-proton sulfur substitution but without radical sulfur chain insertion into the C=C bond. In contrast, there is markedly less H-substitution for the catalytic reaction, even when the reactions are compared for the same degree of polymerization, as judged by development of the 5.4–6.0 and 3.5–4.0 ppm regions (Figs. 4c, 5b), suggesting that the action of the catalyst promotes a greater degree of addition across the double bond in comparison to proton-substitution. However, some degree of proton substitution is still always evident at the early stages of all three reactions (supplementary figures 50–52), suggesting that α-proton substitution may be necessary in activating inverse vulcanization reactions, perhaps aided by the catalyst (supplementary figure 53).

## Discussion

Catalytic inverse vulcanization has been demonstrated. This process is shown to work with a range of catalysts, including low

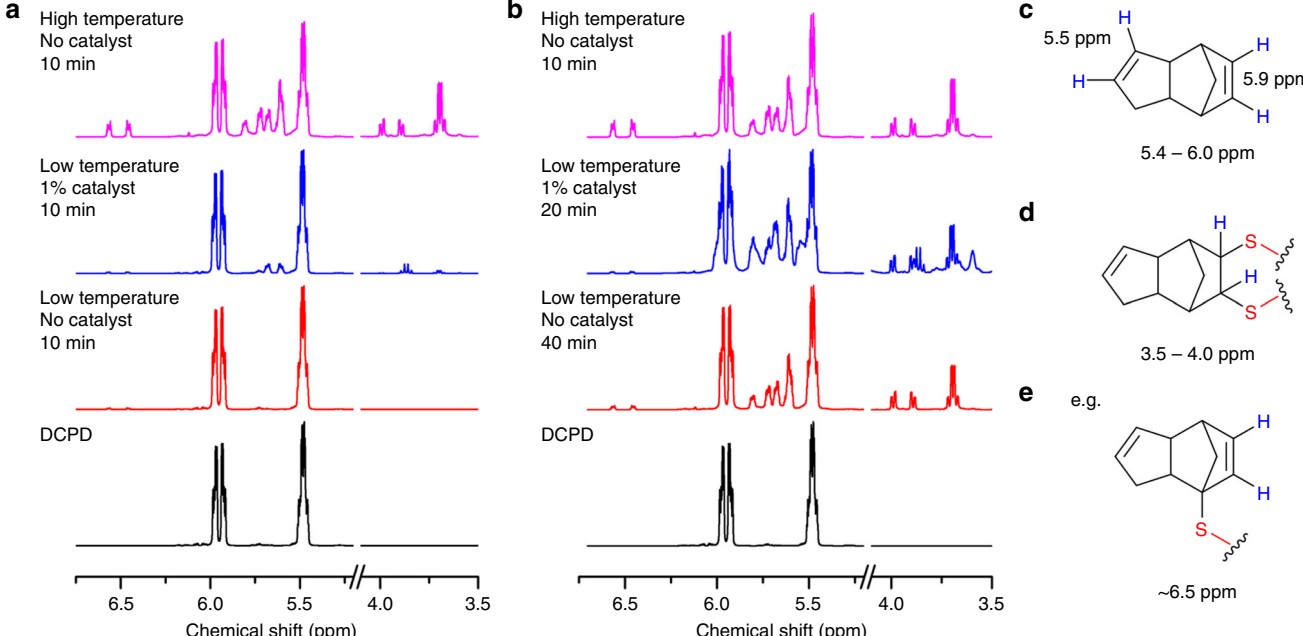

**Fig. 4** Reaction schemes of vulcanizations. **a** Reaction scheme of conventional vulcanization. **b** Reaction scheme for inverse vulcanization (chain transfer, branching, and termination omitted). **c** Suggested catalytic cycle for inverse vulcanization. **d** Representation of the catalyst as a phase transfer agent

**Fig. 5** $^1$HNMR spectra of DCPD at various stages of reaction with sulfur: **a** After 10 min of reaction and, **b** at approximately the same stage of reaction, but under different conditions. Low temperature reactions were carried out at 135 °C, and high temperature at 185 °C. **c** Before reaction, the vinylic protons of DCPD appear at 5.5–5.9 ppm. **d** Reaction with sulfur yields S–C–H protons that appear in the 3.5–4 ppm region. **e** Proton substitution at the α position, without addition across the double bond, results in C═C(–H)–C–S proton environments that are shifted downfield w.r.t. their original positions, appearing at ~6.5 ppm

cost and non-toxic metals. In comparison to un-catalyzed inverse vulcanization, catalysis allows the reaction temperature and time to be reduced, the properties of the polymers to be improved, and the production of dangerous $H_2S$ gas to be significantly inhibited. These factors are likely to greatly enable scale up and use of these fascinating and unique materials. Several unreported thiopolymers have been documented, including from crosslinkers that are unreactive without catalysis. The unlocking of acrylate crosslinker systems significantly increases the number of viable systems. It is hoped that the fundamental chemistry of thiopolymers will be broadened via this catalytic method both by exploration of other cross-linkers, and that future studies on alternative catalysts will be able to improve and optimize their use, as well as elucidating the mechanism further. As an example of this, during publication of this manuscript a related advance has shown that amines can be used to activate the polymerization of sulfur with organic comonomers[58].

## Methods

**Materials**. Sulfur ($S_8$, sublimed powder, reagent grade, ≥99.5%, Brenntag UK & Ireland. Purchased in 25 kg bags), ethylene glycol dimethacrylate (EGDMA, 98%, Alfa Aesar), glyoxal bis(diallyl acetal) (GBDA, Aldrich), *trans,trans,cis*-1,5,9-cyclodo-decatriene (CDDT, 98%, Alfa Aesar), 1,3,5,7-tetravinyltetramethylcyclotetrasiloxane (TVTCSi, 97%, Alfa Aesar), 1,2,4-trivinylcyclohexane (TVCH, 98%, Fluorochem), dicyclopentadiene (DCPD >95%, TCI), 1,3-diisopropenylbenzene (DIB, 97%, Aldrich), divinylbenzene (DVB, 80%, Merck), (*R*)-(+)-limonene (97%, Aldrich), squalene (≥98%, Alfa Aesar), linseed oil (Aldrich), sunflower oil (Tesco PLC, food grade), sodium diethyldithiocaebamate trihydrate (Alfa Aesar), copper diethyldithiocarbamate (TCI), nickel diethyldithiocaebamate (TCI), zinc diethyldithiocarbamate (97%, Aldrich), ZnO (Aldrich), zinc (Aldrich), ZnCl$_2$ (Aldrich), FeCl$_2$ (Aldrich), CuO (Aldrich), CuCl$_2$ (Aldrich), zinc stearate (Aldrich), 2-Cyano-2-propyl benzodithioate (>97%, Aldrich), thiram (Aldrich), chloroform (Aldrich), and chloroform-d (CDCl3, Cambridge Isotope Laboratories Inc.) were commercially available and used as received without any further purification. Iron diethyldithiocaebamate and cobalt diethyl-dithiocaebamate were both synthesized from sodium diethyldithiocaebamate following a method reported in the literature[59].

**Polymerization procedure**. The following is given as a general procedure for the polymerization of sulfur with a crosslinker in the presence or absence of a catalyst. Specific details are given in the supplementary methods section. To a 40 mL glass reaction vial equipped with a magnetic stir bar was added 5 g (19.5 mmol) of elemental sulfur and a catalyst (0, 100, or 500 mg). The reaction was then heated until molten by placing the vial in a metal heating block set to 135 °C. The melting point of sulfur is ~120 °C. The reactions were stirred at 200 RPM using cross shaped magnetic stirrer bars. When the sulfur was molten, 5 g cross-linker was added. The stirring rate was then increased to 900 RPM, and the reaction continued. As the polymerization proceeds the reaction will become first homogeneous (an aliquot removed by spatula will not separate to two phases on cooling), and then solid.

**Characterization**. Gel permeation chromatography (GPC): The molecular weight of the soluble fraction of the polymers was determined by GPC using a Viscotek system comprising a GPCmax (degasser, eluent and sample delivery system), and a TDA302 detector array, using THF as eluent.

Powder X-ray diffraction (PXRD): Data was measured using a PANalytical X'Pert PRO diffractometer with Cu-K$_{\alpha 1+2}$ radiation, operating in transmission geometry.

Differential scanning calorimetry (DSC) were performed on a TA Instruments Q200 DSC, under nitrogen flow, and with heating and cooling rates of 5 °C/min.

Thermogravimetric analysis (TGA) samples were heated under nitrogen to 800 °C at a heating rate of 20 °C min$^{-1}$ using a TA Instruments Q500.

Fourier-transform infrared spectroscopy (FT-IR) was performed using a Thermo NICOLET IR200, between 400 and 4000 cm$^{-1}$. Samples were loaded either neat, using an attenuated total reflectance accessory, or in transmission after pressing into a KBr pellet.

Solution NMR was recorded in deuterated chloroform using a Bruker Advance DRX (400 MHz) spectrometer.

$^{13}$C magic-angle spinning (MAS) NMR spectra were performed on a Bruker Avance III operating at a $^1$H Larmor frequency of 700 MHz, using a Bruker 4 mm HX probe. Chemical shifts were referenced using the CH$_3$ resonance of solid alanine at 20.5 ppm ($^{13}$C). A chemical shielding reference of 189.7 ppm was used, determined from a separate calculation on an optimized tetramethylsilane molecule.

## Data availability

All relevant data that that support the findings of this study are available from the corresponding author upon request.

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

## Acknowledgements

Tom Hasell holds a Royal Society URF. We also thank J. Ellis for elemental analysis, and GC-MS, M. McCarron for GC-MS, S. Moss for ICP-OES analysis, D. Woods and P. Chambon for GPC, and R. Dawson and R. Hanson for TGA. Thanks to J. Chalker, T. McDonald, P. Chambon, and N. Greeves for useful discussions.

## Author contributions

T.H. and X.W. conceived and designed the experiments. The synthetic work was led by X.W. with contributions from B.Z., D.J.P., S.P. and J.A.S. Molding polymers and solution NMR was performed by B.Z. Coating of sulfur-limonene copolymer onto silica and metal uptakes were performed by D.J.P. Solid state NMR was performed by J.M.G. X.W. and T.H. led the writing of the paper with input from all co-authors.

## Additional information

**Competing interests:** The authors declare no competing interests.

