## [Peer Review File · Nature Communications]

Reviewers' comments:

Reviewer #1 (Remarks to the Author):

The manuscript by Wu et al. reports on the use of small molecule accelerators for the polymerization of sulfur with various vinylic and unsaturated comonomers. The work is distinctive by the application of known nucleophilic accelerators from classical rubber vulcanization and related chain transfer agents to afford copolymerizations with a wider scope of monomer. This is a nice advance and the work is well done. This referee is broadly supportive of this work. However, publication in Nature Communications is a stretch given the focus of the work on polymerization chemistry. Hence, this referee recommends publication of this work in a more focused polymer journal (e.g., Macromolecules, Polymer Chemistry, ACS Sustainable Chem&Eng). The only technical concern regards the description of this system as "catalytic." It is clear that a rate enhancement is afforded by the use of these accelerators. But as the authors note in the manuscript, the mechanistic nature of these compounds in classical vulcanization is too complex to assign (this body of literature is very careful to remain ambiguous on whether processes are homolytic vs heterolytic). Hence, it is impossible unlikely that the catalyst is regenerated/retained after polymerization. The recommendation here is to simply rename "catalysis" with a more general descriptor "accelerators", "nucleophilic activators" that doesn't risk incorrect assignment of these processes

Reviewer #2 (Remarks to the Author):

This study by Hasell and co-workers is an exciting advance in the synthesis of polymers from elemental sulfur. In the many recent reports on inverse vulcanization, reaction conditions are typically harsh and require temperatures between 160-190 °C. This limits the types of organic cross-linkers that can be used and also leads to risks of H₂S production and other unwanted side reactions. Addressing this issue, the team discovered a catalyzed version of this reaction, with allows the polymerization to occur below the floor temperature of sulfur. This is an important advance because it allows lower boiling point cross-linkers to be used and it prevents side reactions (H₂S production for example). The more efficient cross-linking also provides improved material properties (such as increased glass transition temperature), with several examples provided by the authors. On top of this, the authors also introduce some technical advances in coating silica with these polymers and then using them in heavy metal sorption. I suspect that this catalyst will be tested and adopted by most groups in this field, so I recommend publication. Below are some comments, questions and suggestions for the authors to consider in a revision.

Suggested revisions and comments:

The proposed mechanism in Figure 4 is understandably speculative because it is without a doubt a complicated process. I actually think it is ok to publish this study without fully understanding the mechanism, which is probably a series of follow-up projects in itself. With that said, here are a few comments for the authors to consider in the revision (especially of figure 4 and the associated discussion):

- i) The reactive phase-transfer mechanism seems plausible and this is a nice way to deal with the immiscibility problem of inverse vulcanization. Can the authors detect the proposed polysulfide by mass spectrometry? This could be tested, for instance, by reacting the catalyst in an equal molar ratio to (molten) sulfur, extracting the product into an organic solvent, and taking a mass spec (ESI, negative mode). This experiment could potentially provide evidence that the catalyst reacts with sulfur, produces an anionic polysulfide as proposed, and also that this species is more soluble than elemental sulfur in organic solvents.
- ii) After the ring-opened S₈ is transferred to the organic phase, the authors propose an ionic reaction with the alkene that invokes the metal as a pi-acid. I think this is too specific a proposal for a few reasons. First, the authors show that the catalyst works for both zinc and sodium as the

cation. Sodium is not a very strong pi-acid, so I don't think direct addition of the sodium thiolate is likely. For non-Michael acceptors like limonene, dicyclopentadiene, and DIB, this ionic addition of thiolate to the pi bond is probably not operative. I suspect a radical mechanism is far more likely for these cross-linkers. This could occur through thermally-induced homolytic scission of an S-S bond in the transferred sulfur-catalyst intermediate, followed by addition to the C=C group. Alternatively, the thiolate could be oxidised to a thiyl radical by air, with subsequent addition to the C=C bond. To test this, can the authors comment on the effect of running the catalyzed reaction under an inert atmosphere? At the very least, the authors should consider removing the ionic addition to a non-electrophilic alkene and including the possibility of a radical addition to the alkene.

iii) For the Michael acceptor alkenes such as EGDMA, the ionic addition to the alkene is reasonable because this would be a classic ionic conjugate addition. The resulting enolate could potentially react with other sulfur species. Have the authors tried running this reaction with a radical inhibitor present? Something like TEMPO or hydroquinone? If the catalyzed inverse vulcanization were purely ionic for EGDMA, then it might work even in the presence of a radical inhibitor. This would also mean that the catalyst can operate by a different mechanism for different monomers.

iv) Basically, the authors need to revise Figure 4 to include the possibility of both ionic and radical mechanisms in the reaction of the transferred sulfur with the alkene. The authors should be conservative and not propose specific structures and arrow-pushing mechanisms without some form of evidence, or at least clearly indicate where intermediates and arrow-pushing is speculative.

In the discussion of mechanism, the authors might also consider noting the chain transfer reactions reported by Pyun in their study on the inverse vulcanisation of styrene: *Journal of Polymer Science, Part A* 2017, 55, 107–116

This is another mechanism by which branching and cross-linking can occur when there are C-H bonds susceptible to H-atom abstraction.

In Figure 5, the authors propose structure iii, which features substitution at the bridgehead carbon. Why is this proposed, rather than substitution at the other (non-bridgehead) allylic position? Bridgehead radicals might be more difficult to form because they cannot planarize.

In Table 1, it might be useful to include a reaction scheme to indicate to the reader exactly what reagents are used and the proposed products

The authors abbreviate diethyldithiocarbamate with "D". This must be changed because D is reserved for deuterium. Readers will see NaD and think it means sodium deuteride, rather than sodium diethyldithiocarbamate. ZnD₂ will also need to be changed for the same reason. Please use a more appropriate abbreviation and replace in both the text, figures and supporting information.

The authors describe sulfur polymers as "self-healing" in the introduction, but I'm not sure this is accurate. Intervention is usually required for repair (e.g. heating the polymer), so it is probably better to describe them as "repairable", "healable" or something similar.

The authors note that vegetable oils have been used as cross-linkers and cite reference 31, which is appropriate, but they should also cite a similar material published by Theato: *Macromol. Chem. Phys.* 2017, 218, 1600303.

In the introduction where the authors mention water purification as an application of sulfur polymers, it may be appropriate to also cite a recent study by Worthington et al on oil-water separation: *Adv. Sustainable Syst.* 2018, 1800024

Using the limonene co-polymer to coat the silica gel is a nice trick that provides a high sulfur, high surface area material. Silica is also used as a hydraulic lubricant for continuous purification of

water, so there is great potential in this discovery for applications in industrial processes and environmental remediation. Is it worth including an image and SEM micrograph of this material as a figure? Also, was less cymene produced when the catalyst was used? This would be consistent with lower amount of produced H₂S.

Reviewer #3 (Remarks to the Author):

I found this paper very interesting, and can see many different ways in which this work can advance. Much of this seems so obvious in hindsight, as many important discoveries do! I found the results very convincing, although some of the mechanistic explanations are not so compelling. However, given how much debate there still is about conventional vulcanization even after many years of investigation this is not very surprising.

My criticisms of the first half of the paper are more stylistic than substantive (see below). I have more technical issues with the mechanistic discussions, but I'm not sure whether this is just because some minor errors have crept into Figure 4c.

Lines 21-23: "Regarding environmental impact and sustainability, it is imperative that access to those synthetic polymers aligns with the principles of green chemistry." Can you re-word it to something that sounds a bit less buzzwordy?

Line 29: "sulfur has been utilized most widely..." -> "Sulfur is widely used..."

Line 30: "... and synthetic rubber via conventional vulcanization" -> "... and vulcanization of natural and synthetic rubbers."

Line 34: probably should be hyphenated - "sulfur-contaminated"

Line 35: "has been interest" - presumably there still is interest?

Figure 1e: I totally understand how it ended up like this, but I am confident you don't actually have any Na²⁺ ions.

Lines 55-60: Very long sentence, poor for readability.

Line 66: I think this is more of an "and" than a "however"

Lines 67-71: Another very long sentence, and slightly confusingly phrased.

Line 72: Not "however" as you are not contradicting the previous sentence.

Line 76: Are you preventing the formation of H₂S, etc, or just minimising it? The former suggests to me that there will be zero H₂S produced.

Line 85: I understand that in technical usage it might not be strictly correct, but if this were my paper I would certainly be trying to get the name "thiopolymers" going, as a less clunky alternative to sulfur-rich polymers!

Line 88: "Catalysts Screening" sounds wrong to my ear. "Catalyst Screening" would be better, but I would go for "Screening of Catalysts"

Line 95: PXRD not defined until line 148, DSC not defined at all.

Lines 99 and 101: You begin two consecutive sentences with "However", which makes me feel like I'm being metaphorically thrown from side to side. My personal rule is no more than one "however" per paragraph!

Line 101: "However, zinc stearate did show some catalytic activity, unfortunately, even after curing there was unreacted or depolymerised sulfur." At first glance I thought it was unfortunate that the zinc stearate showed catalytic activity! I would rephrase it as "Zinc stearate did show some catalytic activity, but unfortunately even after curing there was unreacted or depolymerised sulfur."

Line 166: Very long paragraph

Line 256: In what sense is this a "pre-catalyst"?

Line 265: Though it is speculative, I would suggest the metal might be better than the dithiocarbamate at giving a foothold for the sulfur oligomers to link into before inserting themselves into the catalyst, rather than the ionic/polar nature of the existing bond itself.

Line 267: If the diethyl groups are important at making the catalyst partially oleophilic, why not try the readily available zinc dibutyl dithiocarbamate and see if it is even better?

Line 287: In the case of DCPD, I think your scheme S2 would have to involve a stage with an incredibly high amount of strain on the ring. It seems implausible for me in this case.

Figure 4c: I don't understand how you are getting a di-substituted terminal carbon while also adding a hydrogen to the neighbouring carbon atom, and the mechanism doesn't really try to explain it. Is it supposed to have the sulphur added across the double bond as in figure 4b?

Summary of mechanism section: I think it's fine to explain what you have been able to show about the differences in mechanism between catalysed and uncatalysed, but still be very hand-wavey about the specific mechanism used. The practical aspects are clearly the important part of the paper and they are very good.

Supplemental section: You misspelled "carbamate" as "caebamate", then clearly copy/pasted it repeatedly in the same section. You also don't have experimental methods for the reactions using sodium dithiocarbamate, thiram or the benzodithioate.

David Lowe
Tun Abdul Razak Research Centre

Response to Reviewers comments:

Firstly we would like to thank the reviewers for their time and constructive comments which we believe have improved the manuscript. We have included a revised version, and also a version with tracked changes to allow our modifications to be easily noted.

Reviewers' comments:

Reviewer #1 (Remarks to the Author):

Reply:

We thank the reviewer for the summary that "This is a nice advance and the work is well done. This referee is broadly supportive of this work."

However, we would disagree, in part, with the following statement: "publication in *Nature Communications* is a stretch given the focus of the work on polymerization chemistry."

We agree that the focus of the paper is on polymerisation chemistry. But I would not see this as being an inherent bar to publication in *Nature communications*, which I do not believe restricts itself in this way. Indeed, polymer chemistry has not only previously featured in nature communications, but also *Nature* itself,* as have other aspects of materials chemistry such as metal organic frameworks and covalent organic frameworks, which most people would regard as more 'niche' in focus than polymer chemistry; Especially considering the breadth of use and application of polymers in almost every aspect of modern life.

* For example, published earlier this year: Rapid energy-efficient manufacturing of polymers and composites via frontal polymerization, *NATURE* Volume: 557 Pages: 223+ MAY 10 2018. This paper concerns the polymerisation of DCPD, one of the industrially relevant monomers reported for inverse vulcanisation in the manuscript.

Regarding the one technical concern: "The only technical concern regards the description of this system as "catalytic." It is clear that a rate enhancement is afforded by the use of these accelerators. But as the authors note in the manuscript, the mechanistic nature of these compounds in classical vulcanization is too complex to assign (this body of literature is very careful to remain ambiguous on whether processes are homolytic vs heterolytic). Hence, it is impossible unlikely that the catalyst is regenerated/retained after polymerization. The recommendation here is to simply rename "catalysis" with a more general descriptor "accelerators", "nucleophilic activators" that doesn't risk incorrect assignment of these processes."

We completely understand these concerns of the reviewer, and had considered this nomenclature ourselves. We believe the complex to be working catalytically, and being regenerated, as otherwise it would be unlikely for such small concentrations of the catalyst to allow such significant differences, especially in otherwise unreactive monomers. However, we certainly agree that it would be very difficult to remove the catalyst from the final polymer cleanly, and as such it could be considered to be "used up" in the process and therefore not truly catalytic.

While the process in conventional vulcanisation is commonly referred to industrially as "acceleration" rather than catalysis, there is precedent in the academic literature for it to be described as a catalytic process. For example, the paper "Zinc accelerator complexes. Versatile homogeneous catalysts in sulfur vulcanisation" (*Applied Catalysis A: General* 207 (2001) 55–68).

We would prefer to keep the title and bulk of the text referring to the process as catalytic, as we believe that this is a much more widely understood and accessible term than acceleration – and will be a more appropriate term for the broad readership of a journal such as nature communications. However, we have now added the following section discussing this point and setting out that acceleration may be a technically more appropriate term.:

“When comparable complexes are used to increase the rate of reaction in conventional vulcanisation, they are commonly referred to industrially as “accelerators” rather than catalysts. This mitigates both for the lack of complete understanding of the mechanism, and that it is not possible to separate and reclaim the active complex after polymerisation, as it is incorporated in the product. Here we have used the term catalysis loosely for our process, for sake of accessibility, but acceleration may be more technically appropriate.”

Reviewer #2 (Remarks to the Author):

i) Can the authors detect the proposed polysulfide by mass spectrometry? This could be tested, for instance, by reacting the catalyst in an equal molar ratio to (molten) sulfur, extracting the product into an organic solvent, and taking a mass spec (ESI, negative mode). This experiment could potentially provide evidence that the catalyst reacts with sulfur, produces an anionic polysulfide as proposed, and also that this species is more soluble than elemental sulfur in organic solvents.

Thanks for the good suggestion. In response we have tried mass spec and MALDI with various settings. We were not able to get any useful data from MALDI. We had slightly better luck with mass spec. We tried LC-MS, ESI and negative mode first, as suggested, but unfortunately were not able to resolve signals for any of the samples with this setting, including the catalyst itself. However, on GC-MS, chemical ionisation, positive mode we were able to get some clear signals. We ran tests for: 1) pure sulfur, 2) pure catalyst, 3) catalyst mixed with sulfur and heated to 135 °C. We were able to see clear signals for both the sulfur (as S₈), and the catalyst, in their pure forms. After mixing them together, we only saw signals for the sulfur, and lost the signal for the catalyst. This is likely a result of it reacting with the sulfur on heating (it would not decompose at this temperature – it is below the melting point of the catalyst). Presumably the catalyst reacts with the sulfur to form a higher molecular weight species that was either less susceptible to ionisation, or was above the detection limit of the mass spectrometer (1000 AMU). These data have now been included in the ESI (GC-MS).

ii) After the ring-opened S₈ is transferred to the organic phase, the authors propose an ionic reaction with the alkene that invokes the metal as a pi-acid. I think this is too specific a proposal for a few reasons. First, the authors show that the catalyst works for both zinc and sodium as the cation. Sodium is not a very strong pi-acid, so I don't think direct addition of the sodium thiolate is likely. For non-Michael acceptors like limonene, dicyclopentadiene, and DIB, this ionic addition of thiolate to the pi bond is probably not operative. I suspect a radical mechanism is far more likely for these cross-linkers. This could occur through thermally-induced homolytic scission of an S-S bond in the transferred sulfur-catalyst intermediate, followed by addition to the C=C group. Alternatively, the thiolate could be oxidised to a thiyl radical by air, with subsequent addition to the C=C bond. To test this, can the authors comment on the effect of running the catalyzed reaction under an inert atmosphere? At the very least, the authors should consider removing the ionic addition to a non-electrophilic alkene and including the possibility of a radical addition to the alkene.

We fully agree with the reviewer that it is not possible to say with certainty if it is an ionic or radical mechanism – or indeed a combination of both. In the current manuscript we had stated in the text:

“It is not clear whether this step is radical or ionic in character.”

And

“We tentatively suggest the pathway shown”

We had also tried to keep the figure (4c) ambiguous by showing both a radical ‘dot’ and an ionic ‘dash’ on the sulfur. However, having looked at this figure again, we note that these symbols are perhaps too small to be seen clearly, and we have therefore now enlarged the symbols to make it more explicit that it may be either.

As for some of the monomers used here being non-electrophiles, rubber is a non-electrophile, and as pointed out by the first referee, there is no clear agreement if conventional vulcanisation is radical or ionic.

Regarding the suggestion to try the reaction in an inert atmosphere, we have now performed some additional experiments. These confirm there is no significant difference between running the reactions in an inert atmosphere, or in air – suggesting oxidation by air is not a significant feature. These results have now been included in the ESI. (Tables S7 and S8, and discussion above).

iii) For the Michael acceptor alkenes such as EGDMA, the ionic addition to the alkene is reasonable because this would be a classic ionic conjugate addition. The resulting enolate could potentially react with other sulfur species. Have the authors tried running this reaction with a radical inhibitor present? Something like TEMPO or hydroquinone? If the catalyzed inverse vulcanization were purely ionic for EGDMA, then it might work even in the presence of a radical inhibitor. This would also mean that the catalyst can operate by a different mechanism for different monomers.

iv) Basically, the authors need to revise Figure 4 to include the possibility of both ionic and radical mechanisms in the reaction of the transferred sulfur with the alkene. The authors should be conservative and not propose specific structures and arrow-pushing mechanisms without some form of evidence, or at least clearly indicate where intermediates and arrow-pushing is speculative.

As for the point above, we have now re-drawn this figure to include both charge and radical symbols. The tentative nature of this suggested mechanism is also stated in the text, as is the possibility of it being either ionic or radical in nature.

Following the reviewer's suggestion, we have now performed some of the polymerisations in the presence of TEMPO. We did not see any evidence of inhibition of the polymerisation. This should therefore indicate an ionic route is more likely. However, we are hesitant to draw any strong conclusions from this as the high temperatures required, and high sulfur concentration, may outweigh the inhibiting strength of TEMPO. We have included these new results in the ESI (Table S9, and surrounding discussion).

In the discussion of mechanism, the authors might also consider noting the chain transfer reactions reported by Pyun in their study on the inverse vulcanisation of styrene: *Journal of Polymer Science, Part A* 2017, 55, 107–116

This is another mechanism by which branching and cross-linking can occur when there are C-H bonds susceptible to H-atom abstraction.

A reference to this work has now been added to the discussion of other reports of hydrogen abstraction in inverse vulcanisation.

In Figure 5, the authors propose structure iii, which features substitution at the bridgehead carbon. Why is this proposed, rather than substitution at the other (non-bridgehead) allylic position? Bridgehead radicals might be more difficult to form because they cannot planarize.

We proposed the structure shown in 5c)iii) as accounting for the peaks at 6.5 ppm, which we have attributed to the shift of the peaks for the norbornene double bond (from ~6.0 ppm) upon hydrogen abstraction from the bridgehead position, as stated. We felt this was more likely than these peaks being a shift of the cyclohexane double bond position (at ~5.5 ppm). Further NMR spectra in the ESI (S45 – S51) support this assignment. It is quite possible that there will also be some hydrogen abstraction adjacent to the other (non-bridgehead) allylic position. However, these signals are likely masked under the other peaks in the ~5.75 ppm region, so we would prefer not to speculate.

In Table 1, it might be useful to include a reaction scheme to indicate to the reader exactly what reagents are used and the proposed products

A scheme of S8 +EGDMA reacting to form a crosslinked polymer has now been added to the table.

The authors abbreviate diethyldithiocarbamate with “D”. This must be changed because D is reserved for deuterium. Readers will see NaD and think it means sodium deuteride, rather than sodium diethyldithiocarbamate. ZnD2 will also need to be changed for the same reason. Please use a more appropriate abbreviation and replace in both the text, figures and supporting information.

The abbreviation for diethyldithiocarbamate has now been changed to DTC throughout the paper and ESI including all text and figures.

The authors describe sulfur polymers as “self-healing” in the introduction, but I’m not sure this is accurate. Intervention is usually required for repair (e.g. heating the polymer), so it is probably better to describe them as “repairable”, “healable” or something similar.

A precedent has already been set for calling these polymers “self-healing” in the literature (*Recycling and Self-Healing of Polybenzoxazines with Dynamic Sulfide Linkages*, Mustafa Arslan, Baris Kiskan & Yusuf Yagci, Scientific Reports volume 7, Article number: 5207 (2017)). The authors had only intended to continue the phrasing already set out. However, we do agree with the reviewer with regards to their concern, and are quite happy to amend this. Self-healing in the text has therefore been changed to repair.

The authors note that vegetable oils have been used as cross-linkers and cite reference 31, which is appropriate, but they should also cite a similar material published by Theato: *Macromol. Chem. Phys.* 2017, 218, 1600303.

We thank the reviewer for bringing this oversight to our attention, and have gladly now included this reference.

In the introduction where the authors mention water purification as an application of sulfur polymers, it may be appropriate to also cite a recent study by Worthington et al on oil-water separation: *Adv. Sustainable Syst.* 2018, 1800024

We thank the reviewer for bringing this oversight to our attention, and have gladly now included this reference.

Using the limonene co-polymer to coat the silica gel is a nice trick that provides a high sulfur, high surface area material. Silica is also used as a hydraulic lubricant for continuous purification of water, so there is great potential in this discovery for applications in industrial processes and environmental remediation. Is it worth including an image and SEM micrograph of this material as a figure? Also, was less cymene produced when the catalyst was used? This would be consistent with lower amount of produced H₂S.

We agree with the benefit of including an SEM of the material, one is shown in the ESI in figure S38. However, we have now changed the text in the main paper to make this more obvious, from:

“After coating with a 10 wt.% loading of polymer, the silica gel was a fine free flowing powder, maintaining the same particle size, and without aggregation of the particles (Fig. S38).”

To

“After coating with a 10 wt.% loading of polymer, the silica gel was a fine free flowing powder, maintaining the same particle size, and without aggregation of the particles (photographs and SEM images shown in Fig. S38).”

Regarding the cymene production, we could see no significant difference in the NMR of the product and distillate either with or without catalyst (complex NMR signals, spectra submitted as reviewer only), but we did note a marked reduction in the yield of distillate material. This could be due to the increase in the rate of reaction causing a “trapping” of volatile material in the polymer produced, or from a reduction in the amount of small molecule products (such as cymene).

These yields have now been added to the appropriate experimental section of the ESI, with a note as to their significance.

Reviewer #3 (Remarks to the Author):

I found this paper very interesting, and can see many different ways in which this work can advance. Much of this seems so obvious in hindsight, as many important discoveries do! I found the results very convincing, although some of the mechanistic explanations are not so compelling. However, given how much debate there still is about conventional vulcanization even after many years of investigation this is not very surprising.

My criticisms of the first half of the paper are more stylistic than substantive (see below). I have more technical issues with the mechanistic discussions, but I'm not sure whether this is just because some minor errors have crept into Figure 4c.

Lines 21-23: "Regarding environmental impact and sustainability, it is imperative that access to those synthetic polymers aligns with the principles of green chemistry." Can you re-word it to something that sounds a bit less buzzwordy?

I have tried to re-word this as much as I can to sound less dramatic (i.e. removing imperative), but I am struggling to use terms other than environmental impact, sustainability, and green chemistry, as they best define what I mean. For context, green chemistry is defined by IUPAC as follows:

IUPAC definition:

Green chemistry (sustainable chemistry): Design of chemical products and processes that reduce or eliminate the use or generation of substances hazardous to humans, animals, plants, and the environment. Green chemistry discusses the engineering concept of pollution prevention and zero waste both at laboratory and industrial scales. It encourages the use of economical and eco-compatible techniques that not only improve the yield but also bring down the cost of disposal of wastes at the end of a chemical process.

The sentence has now been reworded as: "The environmental impact and sustainability of any alternative synthetic polymer is therefore important to consider, and should ideally align with the principles of green chemistry.²" However, if the editor or referee have any suggestions on how this could be improved, or specifically which part is problematic, I would be happy to change it following their guidance.

Line 29: "sulfur has been utilized most widely..." -> "Sulfur is widely used..."

Agreed – changed as suggested.

Line 30: "... and synthetic rubber via conventional vulcanization" -> "... and vulcanization of natural and synthetic rubbers."

Agreed – changed as suggested.

Line 34: probably should be hyphenated - "sulfur-contaminated"

Agreed – changed as suggested.

Line 35: "has been interest" - presumably there still is interest?

Agreed – changed as suggested.

Figure 1e: I totally understand how it ended up like this, but I am confident you don't actually have any Na_2^+ ions.

Thanks for spotting this – figure now corrected.

Lines 55-60: Very long sentence, poor for readability.

Agreed – this sentence has been split into two.

Line 66: I think this is more of an "and" than a "however"

Agreed – changed as suggested.

Lines 67-71: Another very long sentence, and slightly confusingly phrased.

Agreed – this sentence has been split into two, and reworded to make it clearer.

Line 72: Not "however" as you are not contradicting the previous sentence.

Agreed – changed as suggested.

Line 76: Are you preventing the formation of H₂S, etc, or just minimising it? The former suggests to me that there will be zero H₂S produced.

Good point, we reduced it to the point we no longer detected it, but as the reviewer points out – I really would not want to claim that absolutely none is produced, it might just be a very low level. We have changed preventing to minimising as suggested.

Line 85: I understand that in technical usage it might not be strictly correct, but if this were my paper I would certainly be trying to get the name "thiopolymers" going, as a less clunky alternative to sulfur-rich polymers!

This is a useful suggestion, we have changed the wording to thiopolymers in the article text.

Line 88: "Catalysts Screening" sounds wrong to my ear. "Catalyst Screening" would be better, but I would go for "Screening of Catalysts"

Changed as suggested

Line 95: PXRD not defined until line 148, DSC not defined at all.

We apologise for this omission. Both terms were defined at first point of use in the main text. Before line 148 PXRD only appeared in a table footnote, and DSC only ever appeared in figure captions. We have now defined both at first point of use in the footnote of table 1. I have left the later definition of PXRD in the main text, in case a reader does not see the small definition in the footnote.

Lines 99 and 101: You begin two consecutive sentences with "However", which makes me feel like I'm being metaphorically thrown from side to side. My personal rule is no more than one "however" per paragraph!

That sounds like a good rule to me. These sentences have been changed to remove both uses. It's a word I overuse.

Line 101: "However, zinc stearate did show some catalytic activity, unfortunately, even after curing there was unreacted or depolymerised sulfur." At first glance I thought it was unfortunate that the zinc stearate showed catalytic activity! I would rephrase it as "Zinc stearate did show some catalytic activity, but unfortunately even after curing there was unreacted or depolymerised sulfur."

Thanks, changed as suggested – this is much clearer.

Line 166: Very long paragraph

This paragraph has now been split into two.

Line 256: In what sense is this a "pre-catalyst"?

We have changed this term to simply catalyst.

Line 265: Though it is speculative, I would suggest the metal might be better than the dithiocarbamate at giving a foothold for the sulfur oligomers to link into before inserting themselves into the catalyst, rather than the ionic/polar nature of the existing bond itself.

This is a useful suggestion, and we have now included the following into that part of the discussion:

“The metal coordinated form of the catalyst is likely more susceptible than the thiram form to the insertion of sulfur oligomers into the catalyst.”

Line 267: If the diethyl groups are important at making the catalyst partially oleophilic, why not try the readily available zinc dibutyl dithiocarbamate and see if it is even better?

This is a very interesting suggestion. In response we have tested the reaction of sulfur with EGDMA in the presence of our normal Zn diethyl dithiocarbamate catalyst, and for comparison also with the same amount of a zinc dibutyl dithiocarbamate catalyst. The results, included here for reviewer only, are as follows:

Different catalysts were tested in reaction of sulfur and EGDMA.

1. Zn (DTC)₂
2. Zinc dibutyl dithiocarbamate

5g Sulfur, 5g EGDMA and 0.1 g catalyst were mixed and reacted at 160 °C.

	Reaction time*/min	T _g /°C
Zn (DTC) ₂	40	19
Zinc dibutyl dithiocarbamate	55	14

* Reaction time was taken as the time when the reaction become dark and homogeneous liquid

There was not a significant enough difference for us to make a strong conclusion. Such a comparison is also hampered by the complexity of the system. For instance – would a more oleophilic catalyst be expected to be better because it transferred into the monomer phase more readily, or worse because it then did not partition into the sulfur phase as well? We think it is worth trying to understand this better, and from this point of view would prefer to leave these inconclusive results out of this paper. Instead, we intend to perform a more in depth further investigation of this. This by nature needs to involve not one, but many changes of the catalyst structure, and multiple repeat reactions. Unfortunately, to get a better understanding (by changing the catalyst more significantly), we will be required to synthesize many of the catalysts from scratch. This will likely take us considerable time. Therefore, while definitely worth further investigation, we believe it would be better left to a future study.

Line 287: In the case of DCPD, I think your scheme S2 would have to involve a stage with an incredibly high amount of strain on the ring. It seems implausible for me in this case.

The referee may well be correct here, this mechanism was suggested by other authors in a previous paper, so we thought it only right to at least discuss this, but have added a comment regarding the strain to the caption of this figure.

Figure 4c: I don't understand how you are getting a di-substituted terminal carbon while also adding

a hydrogen to the neighbouring carbon atom, and the mechanism doesn't really try to explain it. Is it supposed to have the sulphur added across the double bond as in figure 4b?

We thank the reviewer thoroughly for their care and attention in spotting this error. Essentially a 'chemdraw typo' – the reviewer is quite correct, we put the sulfur on the wrong carbons, and have now corrected the figure appropriately.

Summary of mechanism section: I think it's fine to explain what you have been able to show about the differences in mechanism between catalysed and uncatalysed, but still be very hand-wavey about the specific mechanism used. The practical aspects are clearly the important part of the paper and they are very good.

Supplemental section: You misspelled "carbamate" as "caebamate", then clearly copy/pasted it repeatedly in the same section. You also don't have experimental methods for the reactions using sodium dithiocarbamate, thiram or the benzodithioate.

The spelling errors have been corrected, and the experimental details added.

Reviewers' comments:

Reviewer #1 (Remarks to the Author):

The revised manuscript reports on the use of small molecule accelerators, widely used in rubber vulcanization, to polymerize a wider range of comonomers for the inverse vulcanization process with liquid sulfur. This referee previously recommended for rejection of this journal to Nature Communications and publication in a more specialized journal, such as, Macromolecules. That evaluation has not changed. This is simply because the work has a fairly narrow focus in polymer chemistry and builds upon the existing efforts on sulfur polymerizations. However, if the editor wishes to pursue evaluation of this manuscript for review in Nature Commun., then a number of technical concerns (detailed below) must be addressed in an additional revision. Before doing so, there are some non-technical comments that warrant discussion, a these pertain to certain terms chosen by the authors' in this submission.

General Comments

The novelty of this work is the application of known small molecule accelerators for inverse vulcanization processes to expand the scope of monomers/copolymers that can be prepared. This is elegantly conducted piece of work and a well-written manuscript. As the inventor of inverse vulcanization, this referee is certainly able to comment on the benefits and novelty of the findings in this manuscript.

In a general sense, it is very appropriate for the authors of this submission to promote the impact of their findings, particularly, as the corresponding author is a young Academic doing excellent work. However, it is not uncommon in such circumstances for new papers to somewhat besmirch the novelty of earlier work, particularly in this day and age where self-promotion is essential for the success of young scientists. However, in certain spots in the abstract and the introduction, there are clear instances where the assertions of the authors on the limitations of earlier work are subjectively overstated. These have been pointed out in detail below. The larger issue comes in the term "thiopolymer" for sulfur derived polymers in this work. There is already attempts in the literature to rename this class of materials, which is now getting out of hand. Since we developed this new field in 2013 with our Nature Chemistry paper, we referred to these materials simply as a new class of polysulfides, which we later renamed as "Chalcogenide Hybrid Inorganic/Organic Polymers." The polysulfides described in the current revision certainly fall under the general rubric of this class of materials which we initially discovered, so re-naming of this "thiopolymers" is certainly redundant and unnecessary. This concern should be remedied in the current manuscript. Secondly, the developed of alternative low temperature polymerization methods to prepare polysulfides from elemental sulfur, which we termed Dynamic Covalent Polymerizations. This is an alternative solution to the authors findings in this submission, and should be alluded to as achieving such in the Introduction.

Thirdly, independent of the authors' submission, a recent report on small molecule accelerators for inverse vulcanization and dynamic covalent polymerizations which enable a wider scope of vinylic comonomers to be polymerized at lower temperature (*J. Polym. Sci., Part A: Polym. Chem.* 2018, DOI: 10.1002/pola.29266). This should be acknowledged in the introduction as an independent and alternative advance in this field.

Finally, the term catalysis is unsubstantiated in the title and manuscript. Unless the authors can prove that the processes reported are truly catalytic, then an alternative description is required in the manuscript title and throughout the text. This in no way diminishes the importance of the rate acceleration effects observed in this manuscript, but should be done in a fashion that is responsible and correct.

Technical and Detailed Comments

1) In the abstract, lines 8 & 9 are overstated. Replace "harsh reaction conditions" with "elevated temperature" and remove "often-poor properties." It is clear that authors are not being malicious, but these comments are opinionated. If they wish to state facts like (require T's greater than 130C, or have been limited to vinylic comonomers), this would be more appropriate. "Often-poor properties" can be misleading.

2) Lines 53, 60-thiopolymers should be removed as term. For citing of the refractive index, the

following ref must be included (Pyun et al. ACS Macro Lett. 2017, 6, 500) and the value of refractive index corrected to $n = 2.10$

3) At the end of line 89, some description of low temperature synthetic advanced to prepare polysulfides from elemental sulfur via dynamic covalent polymerizations and nucleophilic activation must be included (Pyun et al. Polym. Chem. 2017, 8, 5157-5173; J. Polym. Sci., Part A: Polym. Chem. 2018, DOI: 10.1002/pola.29266

4) Figure 2 has technical content which as presentation issues. Fig 2f should have error bars, or should be removed, since the difference in reported conversion are likely within error of each system. Fig. 2g is only meaningful if conversion/MW of each data point plotted with Tg are comparable.

5) Since extension molding of the new copolymers reported are included, then mechanical properties of these new copolymers should also reported and included in the Supporting Information. This can be limited to the two highest Tg copolymers, but something should be included, on mechanicals, since so much of an emphasis was placed on "improved properties."

Reviewer #2 (Remarks to the Author):

The authors have addressed the issues raised in the first round of reviews and I strongly recommend publication. Making polymers from elemental sulfur is a new and rapidly expanding field and the Hasell lab has made an important advance in delivering a catalysed method for inverse vulcanisation. There are several benefits to this study: the reaction is safer (less H₂S produced), the lower temperature allows for a wider array of cross-linkers (e.g. volatile cross-linkers), there is less energy consumption in the catalysed reaction, and the material properties for known polymers are improved because of increased cross-linking (e.g. increased T_g). Several entirely new polymers are also disclosed in this study (Figure 1). The catalyst will likely be adopted by labs and industries interested in this new world of sulfur polymers.

With that said, I have a couple final suggestions regarding the way the mechanism is depicted in Figure 4c and 4d:

In Figure 4c, the authors have put both a negative charge and a radical on sulfur (S-S^{•-}). The intention was to indicate that the reaction may be either ionic or radical or both. However, as drawn it suggests that a radical anion is formed. This is unlikely and not what the authors intended. I recommend the authors simply draw the sulfur bonded to the metal (S-S-Zn-L) and do not specify anything further in the graphic about whether the reaction is radical or ionic. The discussion in the text is sufficient and appropriately circumspect. Likewise, the radical anion on the ligand L should be removed (just show L bonded to Zn). If the authors really want to put a symbol indicating either an anion or radical, they could use $-/\bullet$ instead of $-\bullet$, but I still think this will be confusing to readers. Finally, I suggest that the authors remove the arrow-pushing notation where the sulfur reacts with the alkene. As drawn, both a full-headed and fish-hook arrow are used, which is confusing as both ionic and radical mechanisms are implied in this single step. Just delete these arrows. Where "Generation of polymers" is written, the authors can instead say "Addition to alkenes and generation of polymers" In this way, the authors are not committed to radical or ionic mechanisms. I also caution against drawing an explicit interaction between the alkene and the Zn in this step because there is no direct evidence for this type of interaction.

In Figure 4d, rather than say the "Catalyst picks up sulfur" perhaps it is more appropriate to write something like "Catalyst reacts with sulfur"

Reviewer #3 (Remarks to the Author):

I am happy to recommend this paper for publication. Below are a few very minor points I noticed while reading through the paper.

Line 23: Chemistry NOT Chamistry

Lines 79, 235: inverse vulcanisation is hyphenated here but not elsewhere

Lines 113, 129, 260: diethyldithiocarbamate should be DTC

Ref 36: JAPS gives a "page number" of 43655

David Lowe

Tun Abdul Razak Research Centre

Response to comments given in green (and after >>), referees comments left in black.

Reviewers' comments:

Reviewer #1 (Remarks to the Author):

The revised manuscript reports on the use of small molecule accelerators, widely used in rubber vulcanization, to polymerize a wider range of comonomers for the inverse vulcanization process with liquid sulfur. This referee previously recommended for rejection of this journal to Nature Communications and publication in a more specialized journal, such as, Macromolecules. That evaluation has not changed. This is simply because the work has a fairly narrow focus in polymer chemistry and builds upon the existing efforts on sulfur polymerizations. However, if the editor wishes to pursue evaluation of this manuscript for review in Nature Commun., then a number of technical concerns (detailed below) must be addressed in an additional revision. Before doing so, there are some non-technical comments that warrant discussion, a these pertain to certain terms chosen by the authors' in this submission.

General Comments

The novelty of this work is the application of known small molecule accelerators for inverse vulcanization processes to expand the scope of monomers/copolymers that can be prepared. This is elegantly conducted piece of work and a well-written manuscript. As the inventor of inverse vulcanization, this referee is certainly able to comment on the benefits and novelty of the findings in this manuscript.

In a general sense, it is very appropriate for the authors of this submission to promote the impact of their findings, particularly, as the corresponding author is a young Academic doing excellent work. However, it is not uncommon in such circumstances for new papers to somewhat besmirch the novelty of earlier work, particularly in this day and age where self-promotion is essential for the success of young scientists. However, in certain spots in the abstract and the introduction, there are clear instances where the assertions of the authors on the limitations of earlier work are subjectively overstated. These have been pointed out in detail below.

The larger issue comes in the term "thiopolymer" for sulfur derived polymers in this work. There is already attempts in the literature to rename this class of materials, which is now getting out of hand. Since we developed this new field in 2013 with our Nature Chemistry paper, we referred to these materials simply as a new class of polysulfides, which we later renamed as "Chalcogenide Hybrid Inorganic/Organic Polymers." The polysulfides described in the current revision certainly fall under the general rubric of this class of materials which we initially discovered, so re-naming of this "thiopolymers" is certainly redundant and unnecessary. This concern should be remedied in the current manuscript.

>> We had previously used the term high-sulfur polymers, which is certainly an accurate description of these materials, and which we have used in our previous 7 publications in this field. This was changed to "thiopolymers" at the suggestion of referee 3. Referee 3 suggested this as a shorter and easier to use term, that still conveys the essential message that these are 1) polymers, and 2) contain sulfur. I was happy to make this suggested change as I was in agreement with referee 3, and found the suggestion useful. We would prefer to continue to use thiopolymer, as it is a shorter and more easily understood term than using the acronym "CHIPS" or the expanded form "Chalcogenide Hybrid Inorganic/Organic Polymers" which we find unnecessary, does not specify the use of sulfur,

and has been largely only adopted by the group that originated this designation, and not the wider literature.

Secondly, the developed of alternative low temperature polymerization methods to prepare polysulfides from elemental sulfur, which we termed Dynamic Covalent Polymerizations. This is an alternative solution to the authors findings in this submission, and should be alluded to as achieving such in the Introduction.

>>The temperature required to polymerise organic monomers with sulfur depends on the monomer, we are aware of the publication mentioned here, which describes the initial reaction of sulfur with a monomer that reacts more readily (styrene), before then in the second step reacting with a second monomer, that would be harder to react initially. From the abstract: *“We describe a new process termed, dynamic covalent polymerization (DCP), where the dynamic S-S bonds in poly(S-rSty) liquid polysulfides were thermally activated to generate sulfur radicals that added to vinylic comonomers to prepare novel terpolymer CHiPs. Using this sequential process we demonstrate the ability to incorporate functional comonomers that were otherwise immiscible with liquid sulfur.”* We note that although only 130 C was required to react the sulfur with the styrene, this only made a low molecular weight oligomeric material rather than a higher molecular weight polymer, and that the next stage of the reaction (with the second monomer) was performed at 170 C – therefore this was not a fully low temperature process.

We had already referenced this paper in our manuscript (previously, reference 45).

However, we have now included an additional reference to it in the introduction, as requested:

“For other co-monomers, such as styrene, 130 °C is enough to form oligomeric material.”³⁷

Thirdly, independent of the authors’ submission, a recent report on small molecule accelerators for inverse vulcanization and dynamic covalent polymerizations which enable a wider scope of vinylic comonomers to be polymerized at lower temperature (J. Polym. Sci., Part A: Polym. Chem. 2018, DOI: 10.1002/pola.29266). This should be acknowledged in the introduction as an independent and alternative advance in this field.

We note this advance, which was submitted for publication two weeks after this manuscript had first been sent out to review. It had therefore not previously been discussed, as it has just now been published as a rapid communication. We believe that an in depth discussion of this new paper, submitted after our own may therefore be inappropriate, but have added the following comment to the conclusions to acknowledge this publication and direct the reader:

“... during publication of this manuscript a related advance has shown that amines can be used to activate the polymerisation of sulfur with organic comonomers.”⁵⁸

Finally, the term catalysis is unsubstantiated in the title and manuscript. Unless the authors can prove that the processes reported are truly catalytic, then an alternative description is required in the manuscript title and throughout the text. This in no way diminishes the importance of the rate acceleration effects observed in this manuscript, but should be done in a fashion that is responsible and correct.

>>We would like to refer the reviewer back to our previous response to this concern as it was raised in the previous set of comments, and we believe we addressed it fully then:

[Previous comments and response start]

Regarding the one technical concern: “The only technical concern regards the description of this system as "catalytic." It is clear that a rate enhancement is afforded by the use of these accelerators. But as the authors note in the manuscript, the mechanistic nature of these compounds in classical vulcanization is too complex to assign (this body of literature is very careful to remain ambiguous on whether processes are homolytic vs heterolytic). Hence, it is impossible unlikely that the catalyst is regenerated/retained after polymerization. The recommendation here is to simply rename "catalysis" with a more general descriptor "accelerators", "nucleophilic activators" that doesn't risk incorrect assignment of these processes.”

We completely understand these concerns of the reviewer, and had considered this nomenclature ourselves. We believe the complex to be working catalytically, and being regenerated, as otherwise it would be unlikely for such small concentrations of the catalyst to allow such significant differences, especially in otherwise unreactive monomers. However, we certainly agree that it would be very difficult to remove the catalyst from the final polymer cleanly, and as such it could be considered to be “used up” in the process and therefore not truly catalytic.

While the process in conventional vulcanisation is commonly referred to industrially as “acceleration” rather than catalysis, there is precedent in the academic literature for it to be described as a catalytic process. For example, the paper “Zinc accelerator complexes. Versatile homogeneous catalysts in sulfur vulcanisation” (Applied Catalysis A: General 207 (2001) 55–68).

We would prefer to keep the title and bulk of the text referring to the process as catalytic, as we believe that this is a much more widely understood and accessible term than acceleration – and will be a more appropriate term for the broad readership of a journal such as nature communications. However, we have now added the following section discussing this point and setting out that acceleration may be a technically more appropriate term.:

“When comparable complexes are used to increase the rate of reaction in conventional vulcanisation, they are commonly referred to industrially as “accelerators” rather than catalysts. This mitigates both for the lack of complete understanding of the mechanism, and that it is not possible to separate and reclaim the active complex after polymerisation, as it is incorporated in the product. Here we have used the term catalysis loosely for our process, for sake of accessibility, but acceleration may be more technically appropriate.”

[Previous comments and response end]

We would re-iterate that many polymer catalysts are not able to be recovered, but are still commonly referred to as catalysts. In addition, there is now more recent literature precedent of this. In the newly published paper that this referee has requested us to include:

J. Polym. Sci., Part A: Polym. Chem. 2018, DOI: 10.1002/pola.29266

The authors of this paper accept in their discussion that their active species is not preserved after the polymerisation, however, from their concluding paragraph: “The sulfur polymer with self-activated comonomer 4-VA could also be used for low temperature DCP to incorporate some low boiling point comonomers that are incompatible with uncatalyzed inverse vulcanization or DCP. These approaches provide a new synthetic and organocatalytic method to activate S8 for

copolymerization processes with functional comonomers at a broader temperature range enabling the preparation of new sulfur materials.”

Technical and Detailed Comments

>>We would like to comment to the editor that although the following 5 of the reviewer’s points appear under the heading of “technical and detailed comments”, we do not believe any of them constitute a significant technical concern with the science or data interpretation. Points 1, 2, and 3 refer to terminology and referencing, rather than the scientific discussion of the results presented. Point 4 addresses the presentation of data. Point 5 requests additional work on a subject outside of the scope or topic of this publication. Further to this, we would refer this reviewer back to their own previous comment that *“The only technical concern regards the description of this system as “catalytic.””*.

1) In the abstract, lines 8 & 9 are overstated. Replace “harsh reaction conditions” with “elevated temperature” and remove “often-poor properties.” It is clear that authors are not being malicious, but these comments are opinionated. If they wish to state facts like (require T’s greater than 130C, or have been limited to vinylic comonomers), this would be more appropriate. “Often-poor properties” can be misleading.

>>We have changed the sentence at issue from:

“However, further development of both the chemistry and applications is handicapped by the restricted choice of cross-linkers, harsh reaction conditions required, and the often-poor properties of the resultant polymers.”

To

“However, further development of both the chemistry and applications is handicapped by the restricted choice of cross-linkers and the elevated temperatures required for polymerisation.”

2) Lines 53, 60-thiopolymers should be removed as term. For citing of the refractive index, the following ref must be included (Pyun et al. ACS Macro Lett. 2017, 6, 500) and the value of refractive index corrected to $n = 2.10$

>>The reference suggested for inclusion, and refractive index listed, are from a polymer made using selenium, hence the higher refractive index than sulfur-polymers. As this manuscript refers to sulfur polymers specifically, we think that to give the refractive index for a selenium polymer would be misleading, as selenium polymers are not the subject of this manuscript.

3) At the end of line 89, some description of low temperature synthetic advanced to prepare polysulfides from elemental sulfur via dynamic covalent polymerizations and nucleophilic activation must be included (Pyun et al. Polym. Chem. 2017, 8, 5157-5173; J. Polym. Sci., Part A: Polym. Chem. 2018, DOI: 10.1002/pola.29266

>>The following comment has been added:

For other co-monomers, such as styrene, 130 °C is enough to form oligomeric material.³⁷

This includes the first requested reference, which had already been included in the paper. The second (which was not published previously) has now been included and discussed in the conclusions section:

“... during publication of this manuscript a related advance has shown that amines can be used to activate the polymerisation of sulfur with organic comonomers.⁵⁸”

4) Figure 2 has technical content which as presentation issues. Fig 2f should have error bars, or should be removed, since the difference in reported conversion are likely within error of each system. Fig. 2g is only meaningful if conversion/MW of each data point plotted with Tg are comparable.

>>W.r.t. Figure 2f, we have now included error bars as requested.

>> W.r.t. 2g, it is not apparent to us from this comment why the graph would be any less meaningful if the MW/conversion of the samples were different. The only difference in these three reactions is the wt.% of catalyst added, and a clear trend in Tg can be seen as a result of that. However, all of these reactions resulted in full conversion and comparable molecular weight, in that all were fully insoluble (crosslinked) – as such no MW could be measured, but there was also no measurable residue of starting materials or low molecular weight species. The figure caption has now been updated to specify that these are insoluble polymers.

5) Since extension molding of the new copolymers reported are included, then mechanical properties of these new copolymers should also reported and included in the Supporting Information. This can be limited to the two highest Tg copolymers, but something should be included, on mechanicals, since so much of an emphasis was placed on “improved properties.”

>> It is the authors intention to investigate the mechanical properties of these polymers in terms of tensile and flexural properties. However, this necessitate an involved study, requiring the design and optimisation of appropriate methods to reliably form uniform dogbone and bar samples (and make the required moulds to produce these). We will also then want to perform multiple measurements in terms of reproducibility, consistency, and optimisation. Currently we have no access to such mechanical testing equipment, but we are reaching out to collaborators to instigate such a study. We think it would be more appropriate therefore to omit such further work from the current manuscript. It seems unnecessary to significantly delay the current manuscript for the sake of something that would be a minor detail in the ESI, and that would not change the findings of this work either way. Rather we would prefer to report such a mechanical properties study when there has been time to conduct it comprehensively, and it could be reported clearly in its own right, where mechanical properties are the topic of the paper.

Reviewer #2 (Remarks to the Author):

The authors have addressed the issues raised in the first round of reviews and I strongly recommend publication. Making polymers from elemental sulfur is a new and rapidly expanding field and the Hasell lab has made an important advance in delivering a catalysed method for inverse vulcanisation. There are several benefits to this study: the reaction is safer (less H₂S produced), the lower temperature allows for a wider array of cross-linkers (e.g. volatile cross-linkers), there is less energy consumption in the catalysed reaction, and the material properties for known polymers are improved because of increased cross-linking (e.g. increased T_g). Several entirely new polymers are also disclosed in this study (Figure 1). The catalyst will likely be adopted by labs and industries interested in this new world of sulfur polymers.

With that said, I have a couple final suggestions regarding the way the mechanism is depicted in Figure 4c and 4d:

In Figure 4c, the authors have put both a negative charge and a radical on sulfur (S-S^{•-}). The intention was to indicate that the reaction may be either ionic or radical or both. However, as drawn it suggests that a radical anion is formed. This is unlikely and not what the authors intended. I recommend the authors simply draw the sulfur bonded to the metal (S-S-Zn-L) and do not specify anything further in the graphic about whether the reaction is radical or ionic. The discussion in the text is sufficient and appropriately circumspect. Likewise, the radical anion on the ligand L should be removed (just show L bonded to Zn). If the authors really want to put a symbol indicating either an anion or radical, they could use $-/\bullet$ instead of $-\bullet$, but I still think this will be confusing to readers. Finally, I suggest that the authors remove the arrow-pushing notation where the sulfur reacts with the alkene. As drawn, both a full-headed and fish-hook arrow are used, which is confusing as both ionic and radical mechanisms are implied in this single step. Just delete these arrows. Where "Generation of polymers" is written, the authors can instead say "Addition to alkenes and generation of polymers" In this way, the authors are not committed to radical or ionic mechanisms. I also caution against drawing an explicit interaction between the alkene and the Zn in this step because there is no direct evidence for this type of interaction.

>>All figure changes implemented as requested.

In Figure 4d, rather than say the "Catalyst picks up sulfur" perhaps it is more appropriate to write something like "Catalyst reacts with sulfur"

>>Changed as suggested.

Reviewer #3 (Remarks to the Author):

I am happy to recommend this paper for publication. Below are a few very minor points I noticed while reading through the paper.

Line 23: Chemistry NOT Chamistry

>>Changed as suggested.

Lines 79, 235: inverse vulcanisation is hyphenated here but not elsewhere

>>The hyphen has been removed.

Lines 113, 129, 260: diethyldithiocarbamate should be DTC

>>Thanks – all corrected.

Ref 36: JAPS gives a "page number" of 43655

>>Updated as requested.

David Lowe

Tun Abdul Razak Research Centre

REVIEWERS' COMMENTS:

Editorial Note: Reviewer #3 was asked to review the authors' responses to Reviewer #1's previous comments.

Reviewer #3 (Remarks to the Author):

I disagree with reviewer 1 as I don't think it is correct to call these chalcogenide polymers. To me it suggests an ionic repeating unit with a chalcogen-derived anion and a (probably metal) cation. The repeating unit is elemental sulphur, so I think it is reasonable to call it a thiopolymer. However, you could call it a chalcopolymer, which would include selenium-based polymers and also nod at the original name given to these materials.

Points 2 and 3 seem to have been dealt with suitably by the authors.

I would prefer to use accelerator over catalyst, but I think the authors note provides sufficient clarity that I wouldn't require them to change it.

The other points seem to have been dealt with adequately.